METHODS AND RESOURCES

# Targeted DamID detects cell-type-specific histone modifications in intact tissues or organisms

Jelle van den Ameele[1,¤a], Manuel Trauner[1,¤b], Eva Hörmanseder[1,¤b], Alex P. A. Donovan[1,2,¤c], Oriol Llorà-Batlle[1,2], Seth W. Cheetham[1,¤d], Robert Krautz[1,¤e], Rebecca Yakob[1,2], Anna Malkowska[1,2,¤c], John B. Gurdon[1,3], Andrea H. Brand [1,2,¤c]*

1 The Gurdon Institute, University of Cambridge, Cambridge, United Kingdom, 2 Department of Physiology, Development and Neuroscience, University of Cambridge, Cambridge, United Kingdom, 3 Department of Zoology, University of Cambridge, Cambridge, United Kingdom

¤a Current address: MRC Mitochondrial Biology Unit and Department of Clinical Neurosciences, University of Cambridge, Cambridge, UK
¤b Current address: Institute of Epigenetics and Stem Cells (IES), Helmholtz Zentrum München, Munich, Germany
¤c Current address: NYU Grossman School of Medicine, New York, USA
¤d Current address: Australian Institute for Bioengineering and Nanotechnology (AIBN), The University of Queensland, Brisbane, Australia
¤e Current address: The Bioinformatics Centre, Department of Biology, University of Copenhagen, Copenhagen, Denmark
* andrea.brand@nyulangone.org

## Abstract

Histone modifications play a key role in regulating gene expression and cell fate during development and disease. Current methods for cell-type-specific genome-wide profiling of histone modifications require dissociation and isolation of cells and are not compatible with all tissue types. Here we adapt Targeted DamID (TaDa) to recognize specific histone marks, by fusing chromatin-binding proteins or single-chain antibodies to Dam, an *Escherichia coli* DNA adenine methylase. When combined with TaDa, this enables cell-type-specific chromatin profiling in intact tissues or organisms. We first profiled H3K4me3, H3K9ac, H3K27me3 and H4K20me1 *in vivo* in neural stem cells of the developing *Drosophila* brain. Next, we mapped cell-type-specific H3K4me3, H3K9ac and H4K20me1 distributions in the developing mouse brain. Finally, we injected RNA encoding DamID constructs into 1-cell stage *Xenopus* embryos to profile H3K4me3 distribution during gastrulation and neurulation. These results illustrate the versatility of TaDa to profile cell-type-specific histone marks throughout the genome in diverse model systems.

## Introduction

Post-translational modifications of histones and chromatin-associated proteins play a crucial role in both normal and pathological cell fate decisions, through their effects on transcription, replication and DNA repair [1]. There remains a need for tools to profile post-translational chromatin modifications in specific cell types, *in vivo*, without prior cell isolation [2]. Most current techniques rely on co-immunoprecipitation (ChIP-seq [3]) or tagging (Cut&Run [4])

**Data availability statement:** Relevant data are within the paper and its Supporting information files and all sequencing data is available from GEO, submission number: GSE278272.

**Funding:** This work was funded by Wellcome Trust Senior Investigator Award (103792), Wellcome Investigator Award (223111) and Royal Society Darwin Trust Research Professorship (RP150061) to AHB. JvdA was supported by a Wellcome Trust Postdoctoral Training Fellowship for Clinicians (105839). EH was supported by a Chromatin Dynamics (CRC1064) and Project Grant (HO 6864/2-1) from the German Research Foundation, Marie Sklodowska-Curie Postdoctoral Fellowship and EMBO Long-Term Fellowship for Postdoctoral Research. JBG and EH were funded by the Medical Research Council (MR/P000479/1). AM is supported by a Wellcome Trust PhD studentship. AHB acknowledges core funding to the Gurdon Institute from the Wellcome Trust (092096) and CRUK (C6946/A14492). The funders had no role in the study design, data collection and analysis, decision to publish, or preparation of the manuscript.

**Competing interests:** The authors have declared that no competing interests exist.

**Abbreviations:** Dam, DNA adenine methyltransferase; IPCs, intermediate progenitors; IUE, in utero electroporation; NSCs, neural stem cells; ORF, open reading frame; RGCs, radial glial cells; TaDa, Targeted DamID.

of DNA associated with the modified histones, and cell sorting is required to obtain cell-type-specific information. More recently, ChromID was developed using histone modification specific biotinylated chromatin reader domains fused to GFP, allowing histone modifications to be profiled by ChIP using antibodies against biotin [5]. Several drawbacks of these approaches include: cross-linking which is biased towards specific regions of chromatin [6]; cell dissociation, which may cause epigenetic changes [7–9]; immunoprecipitation, which can be difficult in early-stage *Xenopus* embryos as they contain large amounts of cytoplasmic protein as compared to the nucleus.

DamID relies on the recruitment of an *Escherichia coli* DNA adenine methyltransferase (Dam) to specific loci in the genome, where it methylates adenines in nearby GATC-motifs [10,11] (Fig 1A). By expressing Dam fused to a chromatin-binding protein of interest, the protein-binding sites are marked by $G^{m6}ATC$ methylation and can be detected by DamID-seq [12]. Cell-type-specific methylation can be achieved through Targeted DamID (TaDa)

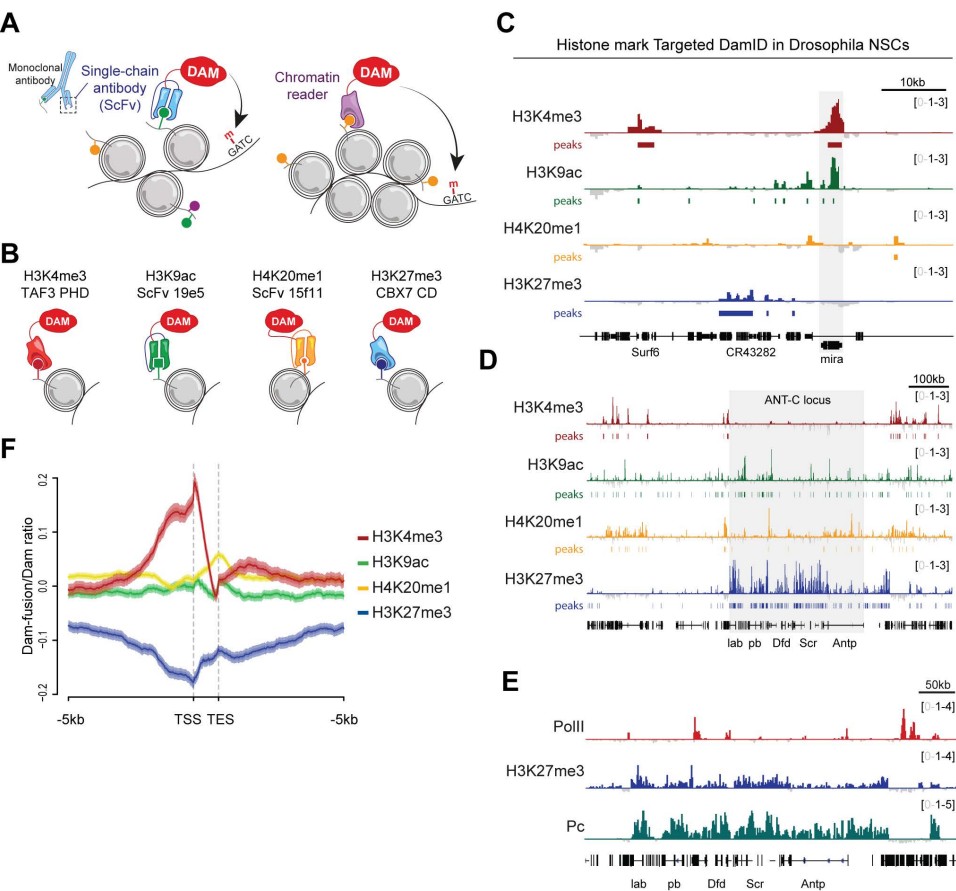

**Fig 1. Profiling histone marks with TaDa.** (A) Schematic showing methods used for Targeted DamID profiling of histone modifications. (B) Illustration demonstrating fusion of Dam to chromatin-binding domains for H3K4me3 or H3K27me3 and to mintbodies (scFvs) against H3K9ac or H4K20me1. (C) Histone marks near the *Mira* locus (shaded) in *Drosophila* third-instar NSCs. Signal files in bigwig format for (**C–F**) are available as supplementary under GSE278272, with the following filename prefixes: H4K20me1: GSE278272_15f11_2-vs-Dam_1, H3K9ac: GSE278272_19e5_1_WOR61-vs.-Dam_1, H3K27me3: GSE278272_CBX7_1-vs-Dam_1 and H3K4me3: GSE278272_TAF3_3-vs.-Dam_3. (D) Histone marks near the *ANT-C* locus (shaded) in *Drosophila* third-instar NSCs. (E) TaDa profiles for PolII, H3K27me3 and polycomb (Pc) around the Antennapedia complex. (E) TaDa profiles for PolII, H3K27me3 and polycomb (Pc) around the Antennapedia complex. (F) Intensity of TaDa signal across genes (transcription start/end site, TSS/TES, ±5 kb) expressed in third-instar larval NSCs. All sequencing files are available at GSE278272.

[13], where the coding sequence of the Dam-fusion protein (ORF2) is inserted downstream of a primary open reading frame (ORF1), thereby reducing the level of Dam expression and avoiding potential toxicity [13]. TaDa has been used to profile chromatin states, where Dam is fused to proteins that recognize and bind specific histone modifications [14]. More recently, DamID has been used for single cell profiling of a subset of histone marks (H3K27me3, H3K4me3, H4K20me1 and H3K9ac) using either entire chromatin-binding proteins, their histone mark-binding domains, or histone modification-specific mintbodies (single-chain variable fragments, scFvs) bound to Dam methylase (EpiDamID, [15]). However, to assay histone modifications in a cell-type-specific manner EpiDamID must be used in combination with single-cell sequencing, due to expression from a ubiquitous promoter.

Here, we describe TaDa for profiling cell-type-specific histone marks in intact organisms and demonstrate its efficacy in *Drosophila* neural stem cells (NSCs) in the developing central nervous system, in NSCs and postmitotic neurons in the developing mouse cerebral cortex and in *Xenopus* embryos. We adapted Dam fusions of chromatin domains for H3K4me3 and H3K27me3 and mintbodies recognizing H4K20me1 and H3K9ac for use with the TaDa protocol, enabling us to profile a variety of chromatin modifications *in vivo*. A Cbx7-Dam fusion recognizes H3K27me3, a mark of Polycomb-dependent facultative heterochromatin found at promoters and gene bodies of highly repressed developmentally regulated genes. In contrast, TAF3-Dam recognizes H3K4me3, which is enriched at active promoters [1]. H3K9 acetylation, recognized by one of the mintbodies, also marks active chromatin and is associated with actively elongating genes, usually close to the 5′ end [16]. Finally, even though previously thought to maintain chromatin compaction, H4K20me1 has been recently shown to promote accessible chromatin at transcribing genes [17]. Thus, we have expanded the toolset of TaDa for profiling chromatin in vivo.

## Results

### Profiling histone modifications with TaDa in *Drosophila* NSCs

We fused Dam to proteins that bind with high affinity and specificity to posttranslational modifications, enabling identification of genome-wide distribution of these marks. We took advantage of several previously characterized mintbodies and protein-binding domains (chromatin readers) that recognize histone modifications on native chromatin (Fig 1A). The PHD domain of TAF3 recognizes H3K4me3 [18], the chromodomain of CBX7 binds H3K27me3 [19], and the single chain monoclonal antibody fragments 19E5 [20] and 15E11 [21] recognize H3K9ac and H4K20me1, respectively (Fig 1B). All chromatin readers were previously validated by ChIP-seq or peptide arrays, and the H3K9ac and H4K20me1 mintbodies were optimized to detect histone modifications upon intracellular expression [20,21] and proven compatible with DamID [15].

We fused these chromatin readers to the *E. coli* Dam methylase, and expressed the Dam-fusions specifically in the NSCs of *Drosophila* third-instar larvae using TaDa [13]. Spatiotemporal control of TaDa expression was achieved with the GAL4 system [22] and GAL80$^{ts}$ [23]. All Dam-fusion proteins generated distinct and reproducible methylation profiles (Figs 1C, 1D and S1A). As expected, H3K4me3 was enriched in promoter regions, whereas H3K9ac, H4K20me1 and H3K27me3 covered both promoter and coding sequences (Figs 1C, 1D and S1B). Focusing on genes actively transcribed in NSCs at this stage [14], H3K4me3 was present at both promoters and coding sequences, while H4K20me1was mainly located in coding sequences. H3K9ac was not highly prevalent but, as expected from its role in a switch from transcriptional initiation to elongation [16], was biased towards the 5′ end of the coding sequence. H3K27me3, which is associated with repressed regions, was absent from expressed genes (Fig 1E and 1F).

We then compared the NSC-specific normalized- and averaged-binding profiles of the chromatin readers and mintbody with published TaDa of Pol II, Brahma, Pc, H1 and HP1α [14] to validate the distribution of the profiled histone marks. Pol II and H3K4me3 correlated with one another well, as expected, whereas H3K9ac correlated better with Brahma, usually linked to enhancer regions marked by H3K27ac. This suggests that in *Drosophila* NSCs H3K9ac might also be associated with enhancers rather that the promoters of expressed genes (S1C, S1E Fig). In terms of repressive marks, H3K27me3 normalized signal correlated highly with the Pc and H1 TaDa, indicating that the chromatin reader is correctly recognizing the presence of H3K27me3 (S1F Fig). In fact, this can be clearly observed at loci known to be targeted by Polycomb Group of proteins, such as the Antennapedia complex or *vnd* locus, where H3K27me3 and Pc-binding patterns overlap (Figs 1E and S1D). Overall, we concluded that our strategy can accurately profile histone marks *in vivo*.

### Profiling histone marks *in vivo* in radial glial cells of the developing mouse cortex

Radial glial cells (RGCs) that line the ventricular zone of the developing mouse cortex can be transiently transfected through injection of plasmid DNA into the telencephalic ventricles of a mouse embryo *in utero*, followed by transcranial electroporation (Fig 2A–2C). We fused the chromatin readers for H3K4me3, H3K9ac, H4K20me1 to intron-Dam for transient transfection [25] and expressed them downstream of pHes5, an established RGC-specific promoter [26,27]. We found that Dam fusions generated distinct methylation profiles when transfected *in utero* (Figs 2D, 2E and S2A), as shown by H3K4me3, H3K9ac and H4K20me1 signal present at *Foxg1* locus, which is expressed in the early mouse telencephalon [28]. In contrast, the active marks were lacking at *HoxB* locus, which is normally only expressed in the hindbrain, rather than the anterior brain (Fig 2E) [29].

The genomic distribution of the profiled modifications mimicked the known distribution of these marks, with H3K4me3 enriched at promoters and H3K9ac and H4K20me1 at gene bodies (S2B Fig). Histone mark peaks in mouse NSCs also reflected peak distribution previously identified by ChIP-seq [30] (S2D Fig), validating our approach. H4K20me1 was not studied as part of the ENCODE project [30], but H4K20me1 TaDa signal was enriched in the gene bodies of genes highly expressed in mouse RGCs (S2C and S2D Fig). The promoters of genes that are highly expressed in mouse RGCs were correlated with the presence of activating H3K4me3 and H3K9ac histone modifications (S2C and S2D Fig).

### Cell-type-specific histone marks throughout cortical neurogenesis

During neurogenesis, RGCs in the developing mouse cortex differentiate into intermediate progenitors (IPCs), and postmitotic neurons that migrate to the cortical plate (Fig 2F). To profile histone modifications during neurogenesis, we used a Hes5 promoter-fragment (pHes5 [26,27]) to restrict H3K4me3 TaDa to RGCs and the promoter of the Tuba1a gene (pTα1 [27,31,32]) to restrict H3K4me3 TaDa to IPCs (Fig 2F). To target postmitotic neurons, we expressed Cre recombinase from a NeuroD1 promoter (pND1 [33]) with a floxDam construct [25] (Fig 2F). The combination gave continuous expression of H3K4me3 TaDa after recombination in young postmitotic neurons (Fig 2G). The histone modification profiles were cell-type-specific, as illustrated by identification of H3K4 trimethylation at the promoter of the progenitor-specific gene Neurog2 in RGCs and IPCs, but not neurons (Fig 2H and 2I). Similarly, the promoter of the neuronal gene Dcx was marked by H3K4me3 in IPCs and neurons, but not in RGCs (Fig 2J and 2K). TaDa generated accurate genome-wide profiles of histone modifications from small numbers of cells, including transient progenitor populations like IPCs, without dissociating them from the complex cellular environment of the developing cerebral cortex.

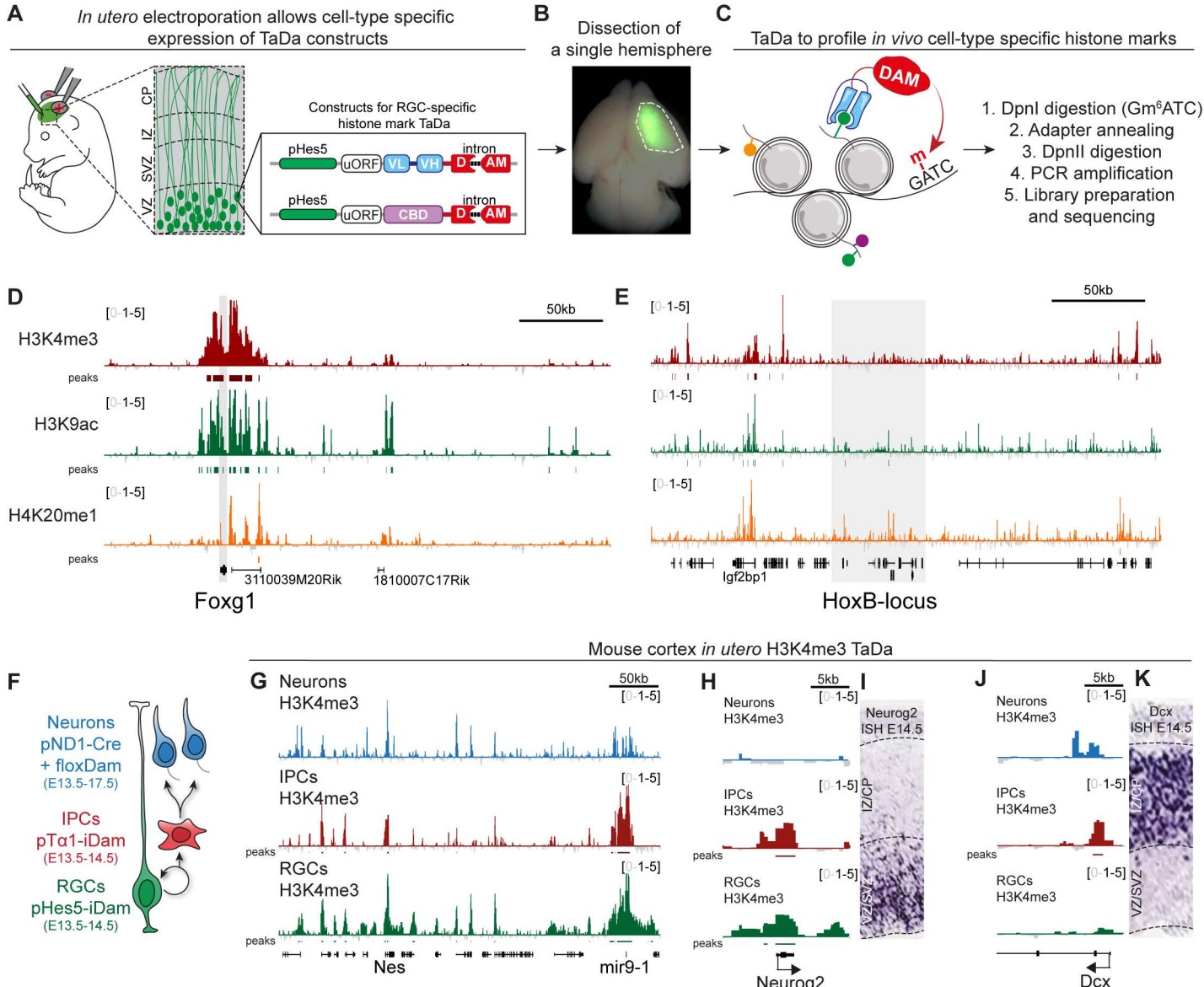

**Fig 2. TaDa in mouse RGCs.** (**A–C**) Schematic of *in utero* TaDa in mouse RGCs. (D) TaDa profiles at *FoxG1* in mouse RGCs. Signal files in bigwig format for (**D–E**) are available as supplementary at GSE278272, with the following filename prefixes: H4K20me1: GSE278272_iue115_5_phes5_15f11nls_5-vs-iue60_i2phes5_dam_S6, H3K4me3: GSE278272_iue62_phes5_taf3-vs-iue53_i2phes5_dam_S2, H3K9ac: GSE278272_iue92_i2phes5_19e5-vs-iue61_i2phes5_dam_S1. (E) TaDa profiles at the *HoxB* locus in mouse RGCs. (F) Schematic of TaDa during mouse cortical neurogenesis. (G) H3K4me3 TaDa profiles at the *Nestin* locus throughout neurogenesis. Signal files in bigwig format for (**G–K**) are available as supplementary at GSE278272, with the following filename prefixes: RGCs: GSE278272_iue62_phes5_taf3-vs-iue53_i2phes5_dam_S2, IPCs: GSE278272_taf3_pta1_iue67_S5-vs-dam_pta1_iue67_S4 Neurons: GSE278272_taf3_flox_iue101-vs-dam_flox_iue72. (**H**, I) H3K4me3 TaDa profiles at the *NeuroG2* locus throughout neurogenesis and the mRNA expression pattern of *NeuroG2* at E14.5. (**J**, K) H3K4me3 TaDa profiles at the *Dcx* locus throughout neurogenesis and the mRNA expression pattern of *Dcx* at E14.5. In situ images in (**I**) and (**K**) were obtained from the Eurexpress database [24]. All sequencing files are available at GSE278272.

## Profiling histone modifications during *Xenopus* embryogenesis

Early development of the *Xenopus* embryo is a powerful system to study cellular differentiation and reprogramming [34] and the role of epigenetic modifications [35]. ChIP experiments from early-stage embryos require large amounts of starting material and have proven difficult due to the yolk and protein-rich cytoplasm. We generated constructs for *in vitro* transcription

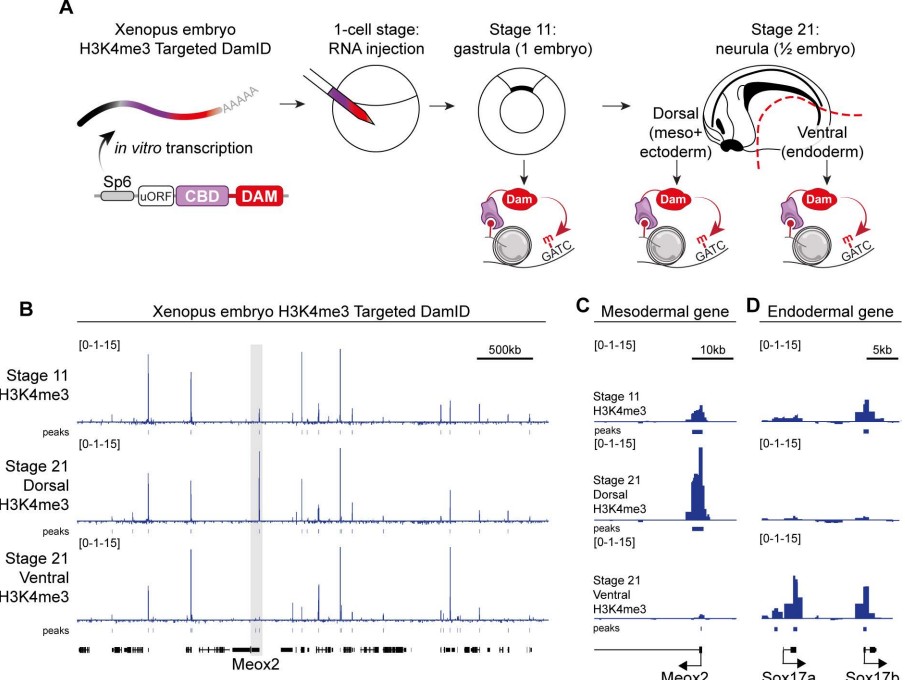

**Fig 3. Cell-type-specific histone mark TaDa in *Xenopus* embryos.** (A) Schematic of H3K4me3 TaDa during Xenopus embryogenesis after injection of in vitro transcribed mRNA in 1-cell stage embryos. (**B**–D) H3K4me3 TaDa profiles at the indicated developmental stages on the *Meox2* and *Sox17* loci. Signal files in bigwig format are available as supplementary at GSE278272, with the following filename prefixes: Stage 11: GSE278272_TAF3_S11_2-vs-Dam_St11_2, Stage 21 ventral: GSE278272_TAF3_S21bot-vs-Dam_S21bot_25, Stage 22 dorsal: GSE278272_TAF3rn_S21top_22-vs-Damrn_S21top_23. All reads files available at GSE278272. All sequencing files are available at GSE278272.

of H3K4me3 TaDa mRNA and injected these into the cytoplasm of fertilized eggs. DamID-seq was performed on stage 11 gastrulae, and on dissected dorsal (ecto- and mesoderm) or ventral (endoderm) halves of embryos at neurula stage 21 (Fig 3A). Despite having only a few embryos as starting material, the H3K4me3 profiles were highly sensitive and specific as demonstrated by selective Dam-methylation on the promoters of genes known to be differentially expressed in mesoderm (*Meox2*; Fig 3B and 3C) or endoderm (*Sox17* locus; Fig 3D). In comparison with ChIP-seq, requiring 50 gastrulae or the tissue from 75 stage 21 embryos [35]), TaDa enabled us to profile H3K4me3 genome-wide in a with high sensitivity in very few embryos.

## Discussion

Here, we show that TaDa can be used to profile posttranslational modifications on histones in a cell-type-specific manner. Recombinant binders are increasingly used in research [36], and chromatin readers have been applied previously to optimise ChIP-seq protocols [18,19,37], or in combination with proximity biotinylation to detect proteins associated with specific histone marks [5]. Chromatin readers have also been used in combination with DamID to allow the profiling of select histone modifications when combined with single-cell sequencing technologies [15]. Combining the versatility of chromatin readers with the strength of TaDa opens new avenues for studying cell-type-specific regulation of chromatin without the necessity for cell dissociation, nuclei extraction or crosslinking. Our results further demonstrate the potential for TaDa to profile any antigen for which a recombinant binder or single chain antibody can

be developed, as we recently demonstrated with nanobodies against GFP [38]. The chromatin readers we tested generated good signals and their distribution across the genome correlated with previously published results. The further development of novel chromatin readers, in particular the optimization of mintbodies for intracellular expression holds great promise in this respect [20,21,36,39].

## Materials and methods

### Plasmids

pHes5, a 764 bp fragment of the mouse Hes5-promoter including the 5'UTR [26], pTa1, a 1097 bp fragment of the mouse Tuba1a promoter [27,31,32], pNeuroD1-Cre and pCAG-Venus were described before [25].

**Dam-fusion proteins.** The H3K4me3 and H3K27me3 chromatin-binding proteins (TAF3 PHD amino acids 856–929 [18], the DNMT3A PWWP domain and CBX7 Chromo Domain amino acids 7–61 [19]) were amplified by reverse transcription (RT)-PCR from cDNA of human embryonic stem cell (ESC)-derived NSCs. DNMT3A PWWP domain-Dam fusion construct was generated for use with TaDa, but not tested. TAF3 PHD was cloned C-terminal, and CBX7 CD N-terminal of Dam. The single chain antibodies against H3K9ac (19e5; gift of Y. Sato [20]), and H4K20me1 (15f11; gift of Y. Sato [21]) were amplified by PCR from plasmids and cloned at the N-terminal Dam. A nuclear localization signal (2xSV40-NLS) was inserted between Dam and all mintbodies or CBX7 CD and Dam.

**Intron-dam constructs.** i1Dam and i2Dam were described previously [25]. i1Dam is intron 3 of mouse IghE [40], placed between the 3rd and 4th helix (BamHI site) of the DNA-binding domain of the Dam methylase [41]. i2Dam is a modified version of the Promega chimeric intron sequence [42–44]. Modifications to the intron and the exon sequence at the splice junctions of i2Dam were made to optimize *in silico* predicted splicing efficiency, remove a weak predicted acceptor site and insert an XmaI site. Splice site predictions were performed with NNSPLICE0.9 [45] and Spliceport [46]. Both constructs were cloned into pCAG-IRES-GFP (pCIG, gift from P. Vanderhaeghen) to obtain pCAG-mcherry-i1/2Dam. Introducing an intron into the Dam CDS has the additional benefit that it greatly improves cloning efficiency, by preventing toxicity in bacteria.

**FloxDam construct.** floxDam was described previously [25] and contains Lox71 and Lox61 sites [47], respectively, upstream of mCherry and within intron 2. The two Lox-sites and intervening sequence are inverted to obtain pCAG-flox2Dam. Cre recombinase activity will result in unidirectional reversion of this cassette, thus reconstituting the i2Dam construct. Placing the Lox-site within the intron has the advantage that splicing will remove the remaining lox-site from the Dam ORF, which otherwise interferes with methylation efficiency [48].

**Xenopus plasmids.** Dam and Dam-TAF3phd were cloned into the EcoRI-NotI sites of pCS2-HA+. mRNA was synthetized in vitro using the MEGAscript SP6 Kit (Ambion, AM1330M) following the manufacturer's instructions.

**Drosophila plasmids.** pUAST-attB-mCherry-i2Dam was cloned by replacing Dam in pUAST-attB-LT3-Ndam [13] with i2Dam from pHes5-mCherry-i2Dam. All Dam-fusion proteins in *Drosophila* were fused to Dam, apart from the CBX7 Chromo Domain, which was fused to i2Dam.

Dam-positive bacteria were 10-beta Competent *E. coli (NEB* C3019H). Dam-negative bacteria were *dam⁻/dcm⁻* competent *E. coli* (NEB C2925H). Plasmids for IUE and transient transfection DamID were prepared from Dam-negative bacteria with Endofree Plasmid Maxi kit (Qiagen 12362).

## Mice and *in utero* electroporation (IUE)

All mouse husbandry and experiments were carried out in a Home Office-designated facility, according to the UK Home Office guidelines upon approval by the local ethics committee, Animal Welfare and Ethical Review Body (AWERB) (Project Licence PPL70/8727). Experiments were done in wild-type MF1 mice. Timed natural matings were used, where noon of the day of plug-identification was E0.5. IUE was performed as previously described [49,50] at E13.5 with 50 ms, 40 V unipolar pulses (BTX ECM830) using CUY650P5 electrodes (Sonidel). DamID plasmids were injected at 1 μg/μl together with pCAG-Venus at 0.25 μg/μl. Embryos were harvested after 24 h (E14.5) for TaDa with pHes5 and pTa1 or after 72 h (E16.5) for TaDa with pND1-Cre.

## Xenopus husbandry and mRNA injection

Mature *Xenopus laevis* males and females were obtained from Nasco. Work with *X. laevis* is covered under the Home Office Project Licence PPL70/8591 after approval by AWERB. Frog husbandry and all experiments were performed according to the relevant regulatory standards, essentially as previously described [35]. For mRNA injection, eggs were in vitro fertilized, dejellied using 2% Cystein solution in 0.1× MMR, pH 7.8, washed three times with 0.1× MMR and transferred into 0.5× MMR for injections. 2–3 pg of mRNA was used per injection. Embryos were cultured at 23 °C and collected at gastrula stage 11 or neurula stage 21. Genomic DNA extraction for DamID library preparation was performed using a Qiagen DNAeasy blood and tissue kit, as per manufacturer instructions.

## Fly husbandry and transgenesis

UAS-mCherry-i2Dam was injected into y,sc,v,nos-phiC31;attP40;+ (Bl#25709) and y,sc,v,nos-phiC31;+;attP2 (Bl#25710) embryos. All other *Drosophila* constructs were only injected into y,sc,v,nos-phiC31;attP40;+ (Bl#25709) embryos. Male transgenic TaDa flies were crossed to w[1118];Worniu-GAL4 (from [51]);Tub-Gal80[ts] (active in NSCs) virgins. Flies were reared in cages at 25 °C, embryos were collected on food plates for 3 hr and transferred to 29 °C for 24 h before dissection at third instar.

## DamID-seq

For *Drosophila* DamID, between 20 and 50 larval brains were dissected. For DamID on mouse cortex after IUE, embryos were cooled on ice, and the electroporated cortex was identified with a fluorescent binocular microscope (Fig 2B). Meninges were removed and the electroporated region microdissected. For Xenopus DamID, five injected embryos were pooled for genomic DNA extraction, and one fifth of the genomic DNA was processed for DamID. All samples were processed for DamID as described previously [12]. DamID fragments were prepared for Illumina sequencing according to a modified TruSeq protocol [12]. All sequencing was performed as single end 50 bp reads generated by the Gurdon Institute NGS Core using an Illumina HiSeq 1500.

## DamID-seq data processing

Analysis of fastq-files from DamID-experiments was performed with the damidseq pipeline script [52] that maps reads to an indexed bowtie2 genome, bins into GATC-fragments according to GATC-sites and normalizes reads against a Dam-only control. Preprocessed fastq-files from mouse, *Drosophila* and *Xenopus* were mapped respectively to the mm10 (GRCm38.p6), dm6 (BDGP6) or Xla.v9.1 genome assemblies. Reads were binned into fragments delineated

by 5′-GATC-3′ motifs (GATC-bins). Individual replicates (see S1 Fig for n-numbers) for the Dam-fusion constructs were normalized against separate Dam-only replicates with a modified version of the damidseq_pipeline [52] (RPM normalization, 300 bp binsize) and all resulting-binding profiles for one Dam-fusion construct were quantile normalized to each other [52]. The resulting logarithmic profiles in bedgraph format were averaged for all GATC-bins across the genome and subsequently backtransformed ("unlog"). Files were converted to the bigwig file format with bedGraphToBigWig (v4) for visualization with the Integrative Genomics Viewer (IGV) (v2.4.19).

### Peak calling and analysis

Macs2 (v2.1.2) [53] was used to call broad peaks for every dam-fusion/dam-only pair on the set of *.bam-files generated by the damidseq_pipeline, using Dam-only as control. Peaks were filtered stringently for $FDR < 10^{-2}$ (mouse) or $FDR < 10^{-5}$ (*Drosophila* and *Xenopus*) and were only considered if present in all pairwise comparisons for a particular experimental condition. Peaks were merged when they were within 1 kb from each other with the merge-function from bedtools (v2.26.0). A similar approach was used for the published ChIP-seq datasets, without control. Associating peaks with the defined genomic features was performed using ChIPseeker (v3.10), with plotAnnoBar function and promoter defined as 2 kb upstream of the transcription start site.

### Genome-wide correlation

Pearson correlation of TaDa replicates (aligned, not normalized reads) in *Drosophila* third instar larval NSCs and of mouse RGC replicates was performed with multiBigwigSummary and plotCorrelation functions from the deepTools suite (v3.5.1). Pearson correlation of signal intensity between previously published NSC TaDa chromatin profiles and signal from Dam fusions generated in this manuscript was performed on bigwig files derived as described in 'DamID-seq data processing' section using the same tools from the deeptools suite. The heat-maps were replotted with the R pheatmap package (v'1.0.12'). Default bin size of 10 kb was used for all correlations.

### Gene coordinates and gene expression data

*Drosophila* NSC list of expressed genes came from [14]. Expressed genes were called from the Pol II DamID-seq bedgraph output using the polii.gene.call R script with default parameters. Mouse RGC gene expression came from [54]. The top or bottom 1,500 genes from the aRG bulk RNA seq dataset (average of 4 replicates) were used for further analysis. Mouse gene coordinates of protein-coding transcripts defined by ensembl were downloaded from biomaRt (v2.38.0). *Drosophila* gene coordinates of "canonical genes" were obtained from the UCSC Table Browser [55].

### Data visualization

Genome browser views were generated using the Integrative Genomics Viewer IGV (v2.4.19) [56] with the midline for TaDa ratio tracks set at 1 and for ChIP-seq set at 0. Peak or gene coordinates were saved in .bed format and supplied as features to Seqplots (v1.12.1) [57]; for S2C Fig quantile normalized, averaged and backtransformed TaDa profiles were provided in bigwig format; for Fig 1F quantile normalized and averaged files in bigwig format were provided. plotAverage and plotHeatmap were used to visualize and average the binding intensities across all supplied coordinates with bin-size of 50 bp. Figures were assembled in Adobe Illustrator.

## Previously published datasets

In situ hybridization images were from the Eurexpress database [24]. Histone mark ChIP-seq datasets from mouse E14.5 forebrain (ENCFF002AME for H3K4me3; ENCFF002ANR for H3K9ac; ENCFF002AFY for H3K27me3) were downloaded respectively from the ENCODE [30] portal as fastq-files and remapped to the mm10 genome assemblies using the just_align function of the damidseq pipeline [52]. *Drosophila* NSC RNA PolII-binding data came from GSE77860 [14]. Mouse RGC gene expression came from GSE65000 [54].

## Supporting information

**S1 Fig. TaDa of histone modifications in *Drosophila* third-instar larval NSCs.** (**A**) Pearson correlation coefficients of aligned reads between all TaDa replicates from *Drosophila* third-instar larval NSCs and Dam-only conditions. (**B**) Genomic feature distribution of TaDa profiles in *Drosophila* third-instar larval NSCs. (**C**) TaDa profiles in *Drosophila* third-instar larval NSCs at the *asense* locus (shaded). (**D**) TaDa profiles in *Drosophila* third-instar larval NSCs at the *vnd* locus (shaded). (**E–F**) Pearson correlation coefficients of normalized and averaged signal files (bigwig) between TaDa using chromatin readers and previously profiled chromatin-binding proteins (Marshall and Brand, 2017). All sequencing files are available at GSE278272 or referenced publication.
(TIF)

**S2 Fig. TaDa of histone modifications in mouse RGCs.** (**A**) Pearson correlation coefficients of aligned reads between all mouse RGC TaDa replicates and the Dam-only conditions. (**B**) Genomic feature distribution of histone mark TaDa profiles in mouse RGCs. (**C**) Average signal intensity (±s.e.m.) of TaDa signal at genes (TSS to TES ±5 kb) expressed in mouse RGCs. Signal files in bigwig format are available as supplementary at GSE278272, with the following filename prefixes: H4K20me1: GSE278272_iue115_5_phes5_15f11nls_5-vs-iue60_i2phes5_dam_S6, H3K4me3: GSE278272_iue62_phes5_taf3-vs-iue53_i2phes5_dam_S2, H3K9ac: GSE278272_iue92_i2phes5_19e5-vs-iue61_i2phes5_dam_S1. (**D**) TaDa (top) and ChIP-seq (ENCODE) profiles for histone modifications near the *Nrarp* genomic locus (shaded). All sequencing files are available at GSE278272.
(TIF)

## Acknowledgments

We thank Y. Sato, S. Koide, P. Vanderhaeghen and W. Gruhn for plasmids and ESC-derived NSC cDNA; K. Harnish (Gurdon Institute NGS Core facility) for help with sequencing; the Gurdon Institute animal facility; C. Bradshaw for advice on bioinformatic analysis, and members of the Brand lab for comments on the manuscript.

## Author contributions

**Conceptualization:** Jelle van den Ameele, Eva Hörmanseder, Alex P. A. Donovan, Oriol Llorà-Batlle, Seth W. Cheetham, Robert Krautz, Anna Malkowska, John B. Gurdon, Andrea H. Brand.

**Data curation:** Robert Krautz.

**Formal analysis:** Jelle van den Ameele, Eva Hörmanseder, Alex P. A. Donovan, Oriol Llorà-Battle, Robert Krautz, Anna Malkowska, Andrea H. Brand.

**Funding acquisition:** Jelle van den Ameele, Eva Hörmanseder, Seth W. Cheetham, Anna Malkowska, John B. Gurdon, Andrea H. Brand.

**Investigation:** Jelle van den Ameele, Manuel Trauner, Eva Hörmanseder, Alex P. A. Donovan, Oriol Llorà-Battle, Seth W. Cheetham, Rebecca Yakob, Anna Malkowska, John B. Gurdon, Andrea H. Brand.

**Methodology:** Jelle van den Ameele, Eva Hörmanseder, Seth W. Cheetham, Robert Krautz, Andrea H. Brand.

**Project administration:** Eva Hörmanseder, John B. Gurdon, Andrea H. Brand.

**Resources:** John B. Gurdon, Andrea H. Brand.

**Supervision:** Eva Hörmanseder, John B. Gurdon, Andrea H. Brand.

**Validation:** Alex P. A. Donovan, Oriol Llorà-Battle, Anna Malkowska, Andrea H. Brand.

**Visualization:** Jelle van den Ameele, Alex P. A. Donovan, Oriol Llorà-Battle, Andrea H. Brand.

**Writing – original draft:** Jelle van den Ameele, Alex P. A. Donovan, Oriol Llorà-Battle, Andrea H. Brand.

**Writing – review & editing:** Eva Hörmanseder, Alex P. A. Donovan, Oriol Llorà-Battle, Seth W. Cheetham, Robert Krautz, Anna Malkowska, John B. Gurdon, Andrea H. Brand.

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
