## [Editor Report · Decision Letter 0]

15 Apr 2024

Dear Andrea,

Thank you for submitting your manuscript entitled "Targeted DamID Detects Cell Type Specific Histone Modifications in vivo" for consideration as a Methods and Resources by PLOS Biology.

I have now had a chance to discuss your manuscript with an academic editor with relevant expertise, and I am writing to let you know that we would like to send your submission out for external peer review.

Once your full submission is complete, your paper will undergo a series of checks in preparation for peer review. After your manuscript has passed the checks it will be sent out for review. To provide the metadata for your submission, please Login to Editorial Manager (https://www.editorialmanager.com/pbiology) within two working days, i.e. by Apr 17 2024 11:59PM.

Kind regards,

Luke

Lucas Smith, Ph.D.

Senior Editor

PLOS Biology

lsmith@plos.org

---

## [Decision Letter · Decision Letter 1]

22 May 2024

Dear Dr Brand,

Thank you for your patience while your manuscript "Targeted DamID Detects Cell Type Specific Histone Modifications in vivo" was peer-reviewed at PLOS Biology. It has now been evaluated by the PLOS Biology editors, an Academic Editor with relevant expertise, and by several independent reviewers. In light of the reviews, which you will find at the end of this email, we would like to invite you to revise the work to thoroughly address the reviewers' reports.

As you will see below, the reviewers report that the tool developed here will be potentially useful for the field, however they raise a number of important concerns and suggestions to strengthen the study further. After discussing these with the Academic Editor, we think many of the reviewer comments will be relatively straightforward to address, although some points may require additional analyses to show the specificity, temporal resolution, and feasibility of the targeted DamID approach. With that said, we do think reviewer 3's major concern and reviewer 1's point 4 may be less straightforward, and we note that these would need to be thoroughly addressed before we can consider your study for publication. Specifically, we think more work will be needed to map H3K27me3 in Xenopus and show that the TaDa tracks indeed reflect HeK27me3 detection. We understand that this may take a substantial amount of effort, but we think this would be essential to validate the method.

Given the extent of revision needed, we cannot make a decision about publication until we have seen the revised manuscript and your response to the reviewers' comments. Your revised manuscript is likely to be sent for further evaluation by all or a subset of the reviewers.

**IMPORTANT - SUBMITTING YOUR REVISION**

*Re-submission Checklist*

*Published Peer Review*

*PLOS Data Policy*

*Blot and Gel Data Policy*

Sincerely,

Luke

Lucas Smith, Ph.D.

Senior Editor

PLOS Biology

lsmith@plos.org

REVIEWS:

Reviewer #1: In their study, van den Ameele et al. expand their previously developed TaDa technique to profile histone modifications genome-wide, in a cell-type specific manner. The great tool developed by the authors allows to track histone mark patterns in cell types in which conventional approaches like ChIP-seq or DamID-seq are not easily performed. They probe the pattern of several histone marks: H3K4me3, H3K27me3, H3K9Ac and H4K20me1; in different biological systems: drosophila neural stem cells, mouse radial glial cells and xenopus embryos. This approach will enable profiling of histone marks or other chromatin factors in a wide range of tissues or cell types. However, this manuscript would benefit from several clarifications, especially about data analysis and data visualization.

My comments to improve this manuscript are:

Major points:

1) Figure legends should be more specific and describe in more details the corresponding panels. Some key information seems missing, such as:

a. A description of the clustering done in Figure 1F.

b. A description of the color code in Figure S2E. Related to that panel,

i. How is the NES calculated?

ii. What do the arrows represent?

2) To further show the specificity of their H3K4me3 TaDa signal in Xenopus embryo, the authors may want to correlate their data obtained in the endoderm with those previously obtained by ChIP-seq in the same cells (Hörmanseder et al., 2017).

3) What is the temporal resolution of TaDa? Could H3K4me3 pattern be assessed at earlier time points in Xenopus development? What is the author's opinion on this?

4) Have the authors mapped H3K27me3 in Xenopus embryos? Performing such an experiment may further consolidate the feasibility of their technique in this organism.

5) In Figure 1C-D and Figure S1C-D, the TaDa signal depicted on IGV tracks ranges from of 0 to 3. However, in Figure1E, the y-axis representing "the average signal intensity of TaDa signal across genes" ranges from -0.2 to 0.2. Could the author explain why there is such a difference between these two scales? How can H3K27me3 signal have negative values? A similar plot can be found in Figure S2H, which has a scale ranging from 0 to 2.2.

6) Pearson correlation usually ranges from -1 to 1 (as shown in Figure S1E-F). However, in Figure S1A and S2D the Pearson correlation ranges from 0 to 1 only. Can the author comment on such a scale? For a better visualization of the data, the authors may want to use the same scale and color-code for all heatmaps representing Pearson correlation coefficients.

7) Related to the previous point, the correlation coefficients between H3K4me3, H3K9Ac and H4K20me1 are very different between Figure S1A and S1E. For example, Figure S1E shows a negative correlation of -0.13 between H3K9Ac and H3K4me3 while Figure S1A shows a positive correlation of ~0.60 between those same marks. What could be the explanation for that?

8) In the Xenopus TaDa section, the authors claim that they map histone modifications in single embryos (last sentence of the paragraph: "TaDa enabled us to profile H3K4me3 genome-wide in a single embryo"). However, it is specified in the methods (DamID-seq section) that five embryos were pooled for genomic DNA extraction. As a consequence, the TaDa technique does not map histone modifications in single embryos but rather in a limited number of embryos. The authors need to tone down their conclusion of this section.

9) To improve the readability of the paper, the introduction should be extended, including more details about the chromatin marks investigated (e.g. their localization in the genome, their role in gene regulation, etc.).

10) The text does not refer to Figures 2D-E, S2A-D or S2G. Please remove those panels from the figures or cite them in the text and reformat the figures accordingly.

Minor points:

1) Figure S1A: a couple of numbers are missing in the matrix

2) Figure S2F: the track labels are misaligned with the tracks themselves.

3) It is mentioned in the first paragraph of the Results that the chromatin mark H3K36me3 was profiled by TaDa. Those data are missing from the Figures, are those data not reliable (as mentioned for H3K9me3 TaDa signal)? Can the authors clarify this point?

4) In the Methods, "DamID-seq data processing" section: the order of the species and their corresponding reference genome is not correct.

5) For a better comprehension and readability of the manuscript, the authors may want to explain in more details their intronDam and floxDam tools, and their use in the mouse brain.

Reviewer #2: The manuscript "Targeted DamID detects cell-type specific histone modifications in vivo" describes the profiling of multiple histone modifications using the authors' previously published targeted DamID (TaDa) method. The author fuse chromatin binding proteins or single-chain antibodies to the Dam adenine methylase Dam in intact tissues to generate cell-type-specific histone modification profiles. The authors apply their method to neural stem cells in Drosophila, neural stem cells in the developing mouse brain, and the 1-cell stage of Xenopus embryos during gastrulation and neurulation. The authors achieve cell-type specificity using either a cell-type specific promoter for the TaDa construct or injecting in vitro transcribed mRNA of the TaDa construct Xenopus embryos. A group recently used DamID bound to mintbodies to detect histone modifications in vivo (PMID: 35366395) but required single-cell sequencing to investigate histone modifications in a cell-type-specific manner while the present manuscript describes a method for achieving cell-type-specificity without requiring single-cell sequencing.

The ability to profile histone modifications in vivo would be of high use to the chromatin community, especially since TaDa is a popular method for profiling in vivo chromatin proteins in Drosophila, and the current manuscript describes a method for effective use in three model organisms. I have only minor concerns.

1). The introduction, discussion, and methods section mention using the PWWP domain of DNMT3A to profile H3K36me3 and a single-chain antibody to H3K9me3. Yet, data for neither modification appears in the manuscript. Please remove mentions of these histone modifications from the text or show data.

2). There appear to be peak annotations beneath all genome browser tracks in all the figures. Please indicate the peak track in the figure or at least in the legend for the figure.

3). Please add a label to the y-axis of Figure 1E to indicate the units.

4). In Figure 1F, there is not designation of what C1-C12 on the heat maps indicate. Please correct.

5). The method for generating the images in Figures 2I and 2K is missing from the Methods section; please correct.

6). Panels A, B, and C of Figure S2 are not mentioned in the text; please correct.

Reviewer #3: In the submitted manuscript, the authors present an extension and validation of their previously published protocol of cell-type specific targeted DamID. DamID takes advantage of a bacterial methyltransferase that, when directed to specific areas of the genome via fusion partners, will create m6A residues at GATC motifs. In this manuscript, the authors demonstrated a system based on cell-type specific promoters to expression fusion proteins and characterize several chromatin marks within drosophila NSCs, mice radial glia, and xenopus embryos. This modification to the protocol is a logical next step yet is impactful and a significant progression from prior work, and the data presented are exciting, with one notable exception that I think needs to be addressed before publication.

Major (essential) issues:

1. I am unconvinced that the H3K27me3 profiling data presented in the paper is working in mouse cells. The data from the drosophila NSCs (Fig 1C) look as I would expect an H3K27me3 profile to look. The data presented in figure 2D and 2E does not qualitatively look like H3K27me3 profiles, however. Instead, they look like the other presented tracks of activating histone modifications. Especially troubling is the divergence between the tracks presented in figure S2F - in general the TaDa tracks look like the ChIPseq tracks, with the exception of the H3K27me3. Additionally, in the correlation matrix shown in Figure S2D, H3K27me3 has a very high correlation with both H3K4me3, but also with the unfused Dam control. Figure S2H also does not look like figure 1E, and the enriched sets for H3K27me3 in figure S2E are mostly activating marks such as H3K27ac and H3K4me3 rather than mostly H3K27me3 sets. In the text, these data are presented as perhaps evidence of bivalent chromatin, but I think these correlations are too high; to me this suggests the fusions may not be working here, and what is actually being measured in the H3K27me3 is something like accessibility (hence the high correlation with the unfused Dam in figure S2D) rather than interactions with the chromatin state. Given its high correlation with the unfused Dam, it is also possible that the H3K4me3 is also simply showing accessible chromatin, which would be less noticeable given the expected correlation between active promoters and accessible chromatin. This should also be checked especially given that H3K9ac doesn't show this correlation.

Minor issues:

1. In figures S1B and S2G, the inclusion of an additional bar depicting the genomic (unenriched) proportion of each of the genomic features, would enhance interpretability.

2. In figure S2F, the labels of the tracks are vertically misaligned.

3. Given that DamID is resolution-restricted to GATC motifs, the early data processing steps are described as occurring in GATC-bounded bins, later analysis occurs in genomic coordinates. Does it make sense to stay in bin-space rather than genome-space for, for instance, peak calling? The methods mention that peaks are merged when within 1kb, but it seems like the local GATC density could affect peak merging here, if for instance a mid-peak GATC-bounded fragment is too large (in nucleotide width) for accurate illumina cluster generation and is artificially lowly-represented. Stitching over bin distance rather than nucleotide distance may alleviate some of this issue? I don't think this is an issue for any analysis presented within the manuscript, but I would be curious how this would affect, for instance, figure 1F.

4. What do the clusters in figure 1F represent? It seems like the biggest correlation (for, say, clusters 3,4,6,9,10,11) is with horizontal position of H3K4me3 related to the TSS? Is there a biological significance there, or is this an artifact of something like local GATC density?

---

## [Decision Letter · Decision Letter 2]

13 Sep 2024

Dear Dr Brand,

Thank you for your patience while we considered your revised manuscript "Targeted DamID Detects Cell Type Specific Histone Modifications in vivo" for publication as a Methods and Resources at PLOS Biology. This revised version of your manuscript has been evaluated by the PLOS Biology editors, the Academic Editor and the original reviewers.

Based on the reviews, we are likely to accept this manuscript for publication, provided you satisfactorily address the remaining points raised by reviewer 1. Please address all concerns from Reviewer 1 regarding the statistics. However, note that we don't require additional H3K27me3 data. Please also make sure to address the following data and other policy-related requests.

1) Please change your title to: "Targeted DamID Detects Cell Type Specific Histone Modifications in intact tissues or organisms"

2) Please modify your ethics statement in the manuscript to confirm that the mouse and Xenopus experiments were specifically reviewed and approved by an Institutional Animal Care and Use Committee (IACUC) and give the full name of the IACUC/local ethics committee that approved it.

3) Please adhere to the PLOS Data policy:

(A)You may be aware of the PLOS Data Policy, which requires that all data be made available without restriction: http://journals.plos.org/plosbiology/s/data-availability. For more information, please also see this editorial: http://dx.doi.org/10.1371/journal.pbio.1001797

-Supplementary files (e.g., excel). Please ensure that all data files are uploaded as 'Supporting Information' and are invariably referred to (in the manuscript, figure legends, and the Description field when uploading your files) using the following format verbatim: S1 Data, S2 Data, etc. Multiple panels of a single or even several figures can be included as multiple sheets in one excel file that is saved using exactly the following convention: S1_Data.xlsx (using an underscore).

-Deposition in a publicly available repository. Please also provide the accession code or a reviewer link so that we may view your data before publication.

Regardless of the method selected, please ensure that you provide the individual numerical values that underlie the summary data displayed in the following figure panels as they are essential for readers to assess your analysis and to reproduce it: Figure 1F, S2C

(B) Please deposit the raw sequencing files for the TaDa profiles in the GEO or similar data repository. Please ensure that the deposition is made publicly available at this stage and provide the accession number/weblink in the Data Availability Statement in the online submission form. We require the sequencing data for the following figures:

Figure 1C-E, 2D-E, 2G-H, 3B-D, S1C-D, S2D

We expect to receive your revised manuscript within two weeks.

*Published Peer Review History*

*Press*

Sincerely,

Suzanne

Suzanne De Bruijn, PhD,

Associate Editor

sbruijn@plos.org

PLOS Biology

Reviewer remarks:

*Reviewer #1:

In their revised version of the manuscript, the authors addressed most of the points raised by the reviewers. However, the authors only partially addressed the major point raised by the editors. Indeed, it was specified in the response to the authors that "reviewer 3's major concern and reviewer 1's point 4 would need to be thoroughly addressed before considering the manuscript for publication", and "more work will be needed to map H3K27me3 in Xenopus and show that the TaDa tracks indeed reflect H3K27me3 detection".

The authors addressed reviewer 3's major comment by removing H3K27me3 data generated in the mouse cortex.

Reviewer 1's point 4 was not addressed, the authors did not map H3K27me3 in Xenopus. Thus, H3K27me3 was mapped in Drosophila neural stem cells only.

Another important point raised my concern while assessing the authors' rebuttal.

On their answer to reviewer 1's point 6), the author state that: "Pearson correlation ranges from 0 to 1". However, Pearson correlation coefficient, just like Spearman correlation coefficient, ranges from -1 to 1 (ref 1, 2). It is worrisome and I am now unsure of what is actually plotted in Figures S1A and S2A. How do the authors calculate the correlations? Which package are they using? Maybe the authors are in fact plotting the coefficient of determination, R2, which ranges from 0 to 1?

Along a similar line, the authors state in the answer to reviewer 1's point 7) that: "Pearson correlation 0.6 represents a similar level of correlation to a Spearman value of -0.13". Again, this statement is incorrect. The correlation between Pearson and Spearman correlation coefficients depends on the data type (ref 3). For linear and monotonic data, Spearman and Pearson correlation coefficients should be similar.

These points need to be clarified as they question the authors' comprehension of the statistics used in the manuscript.

Minor comments:

1) Add peak annotation in Figure 1C-D.

2) Please correct the color scale in Figure S1A-E-F and S2A so the white color aligns with the middle of the scale bar (i.e. value of 0,5 in figures S1A and S2A and of 0 in Figure S1E-F).

1: https://en.wikipedia.org/wiki/Pearson_correlation_coefficient

2: Mukaka MM. Statistics corner: A guide to appropriate use of correlation coefficient in medical research. Malawi Med J. 2012 Sep; 24(3):69-71

3: Hauke J., Kossowski T.. Comparison of Values of Pearson's and Spearman's Correlation Coefficients on the Same Sets of Data. Quaestiones Geographicae. 2011; 30(2):87-93.

*Reviewer #2: The authors addressed all of my concerns in the revision and I believe the manuscript is acceptable for publication now.

*Reviewer #3: My concerns have been addressed with the revised submission and the manuscript is suitable for publication.

---

## [Editor Report · Decision Letter 3]

21 Oct 2024

Dear Dr Brand,

Thank you for your patience while we considered your revised manuscript "Targeted DamID Detects Cell Type Specific Histone Modifications in Intact Tissues or Organisms" for publication as a Methods and Resources at PLOS Biology. This revised version of your manuscript has been evaluated by the PLOS Biology editor.

Thank you for addressing many of our concerns, and providing additional information about the GEO dataset. There are a few remaining concerns that I'd like you to address before we can accept your manuscript:

1) Thank you for providing us with the GEO token. However, please note that although we can editorially accept your manuscript without these data, They will need to be publicly accessible before final acceptance, due to our editorial policies.

2) Regarding the raw data underlying Figure 1F and S2C: I appreciate that you provide the underlying bigwig files in the GEO data. However, can you please ensure it is clear which files are underlying these figures? This can be done by relabeling them, or ensuring that this information is stated in the legends.

3) Please ensure that the source of the data is mentioned in the figure legends of all main and supplemental figures. I appreciate that in your case this would be repetitive, but it is editorial policy.

We expect to receive your revised manuscript within two weeks.

*Published Peer Review History*

*Press*

Sincerely,

Suzanne

Suzanne De Bruijn, PhD,

Associate Editor

sbruijn@plos.org

PLOS Biology

---

## [Editor Report · Decision Letter 4]

18 Nov 2024

Dear Dr Brand,

This is Luke Smith -  I have recently taken over the handling of your PLOS Biology submission, "Targeted DamID Detects Cell Type Specific Histone Modifications in Intact Tissues or Organisms" from my colleague, Dr. Suzanne De Bruijn, who was covering for me while I was away on parental leave this summer/fall. I have now had a chance to take a look through this most recent revision, and the changes made in response to our previous editorial requests, and I am pleased to say that, on behalf of my colleagues and the Academic Editor, Tom Misteli, we can in principle accept your manuscript for publication. While we are happy to editorially accept your study, please note that there may be additional requests from our production team, regarding any remaining formatting and reporting issues - and these will need to be addressed before publication. These will be detailed in an email you should receive within 2-3 business days from our colleagues in the journal operations team; no action is required from you until then. Please note that we will not be able to formally accept your manuscript and schedule it for publication until you have completed any requested changes.

**IMPORTANT: As a last note, I took the liberty of updating the data availability statement in our editorial manager system to more closely resemble the statement that you provided in your manuscript (and to be compliant with our data reporting policy). The data availability statement in EM now reads:

"Relevant data are within the paper and its Supporting Information files and all sequencing data is available from GEO, submission number: GSE278272"

Please let me know if you see any issue with this change.

PRESS

Sincerely, 

Lucas Smith, Ph.D.

Senior Editor

PLOS Biology

lsmith@plos.org